# On the Application of Time-Series Foundation Models for Detecting Long-Context Anomalies in Industrial Control Systems

Alexa Lowe [1]   Clement Fung [1]   Lujo Bauer [1]

## Abstract

Machine learning (ML) can be used to protect critical, industrial processes. By predicting future industrial-process values, ML models can identify anomalies indicative of harmful malfunctions or cyberattacks. However, training ML models for industrial processes in practice is hindered by the scarcity of high-quality training data and technical expertise. Thus, we explore the application of pre-trained time-series foundation models (FMs) for detecting anomalies in industrial processes. In an evaluation with two time-series FMs and three ICS, we find that FMs struggle to detect long-duration anomalies, which are common when ICS are attacked. We introduce a new time-series forecasting method that filters out suspicious data and uses previously predicted data as input, called Forecast Fallback (FF). We show that FF significantly improves performance for detecting ICS anomalies (i.e., 0.1–0.45 increase in ROC-AUC). Our work demonstrates challenges for using time-series FMs to detect ICS anomalies and illustrates open challenges for future work that uses time-series FMs for critical, industrial processes.

## 1. Introduction

Industrial control systems (ICS) are commonly used in critical infrastructure, such as water and energy systems (Stouffer, 2011). To attack ICS, an adversary can gain access to the industrial network and inject false sensor readings and commands to disrupt the physical process. These attacks can be executed over long durations (Liu et al., 2011; Ye & Zhang, 2020), with the potential to cause massive impact and harm (Slowik, 2019; Beerman et al., 2023). A common proposal for defending ICS from such attacks is to use machine learning (ML) to detect anomalies in process

data (Fung et al., 2022; Lamberts et al., 2023). This involves training a new ML model for time-series forecasting and requires high-quality training data and ML expertise that is rare in ICS contexts (Fung et al., 2025).

To alleviate the need for training new ML models for each system, we investigate the application of off-the-shelf, time-series foundation models (FMs) for anomaly detection. Time-series FMs have yet to be applied to ICS but have shown promise in a variety of forecasting domains, such as transportation, weather, and web analytics (Ansari et al., 2024). However, ICS anomalies may have properties that make them different from anomalies in other time-series contexts (Wu & Keogh, 2021; Wagner et al., 2023). First, whereas time-series anomalies are often short, ICS anomalies are often *long-context*; adversaries may manipulate the process data subtly for several hours or days. Second, process values (i.e., sensor readings and actuator commands) are highly connected: an attack on one sensor often induces changes across dozens of sensors, and capturing these inter-feature interactions is important to detect ICS anomalies.

To evaluate time-series FMs for ICS anomaly detection, we apply two state-of-the-art multivariate models, Amazon's Chronos-2 (Ansari et al., 2024) and Salesforce's Moirai-2.0 (Liu et al., 2025), to publicly available anomaly data of three ICS: a water treatment system (Goh et al., 2016), a chemical process (Bathelt et al., 2015), and a water distribution system (Murillo et al., 2021). Using traditional anomaly detection methods, We find that FMs perform poorly when faced with long-context anomalies that are prevalent in ICS attacks, and lightweight fine-tuning also does not significantly help. In response, we propose a new forecasting method called Forecast Fallback (FF) that uses a selective combination of new time-series data and previously predicted values in real-time. FF helps FMs avoid incorporating anomalous data into predictions, resulting in improved anomaly-detection performance (i.e., ROC-AUC increases by 0.1–0.45) that matches state-of-the-art, trained ML models for ICS (i.e., F1 score of 0.83 on the CTown dataset). Our work serves as a preliminary investigation of the challenges of using time-series FMs for ICS anomaly detection and motivates new approaches for FM-based forecasting for ICS.

[1]Carnegie Mellon University, Pittsburgh, PA, USA. Correspondence to: Clement Fung <clementf@andrew.cmu.edu>.

*Proceedings of the 2nd ICML Workshop on Foundation Models for Structured Data*, Seoul, South Korea. 2026. Copyright 2026 by the author(s).

## 2. Background

We first describe the anomalies in three most commonly used datasets amid the limited availability of ICS anomaly data (Fung et al., 2025) in Sec. 2.1, followed by an overview of standard detection approaches in Sec. 2.2.

### 2.1. Anomalies in industrial control systems (ICS)

ICS contain several devices: sensors read information from the physical process (e.g., water pressure), controllers use control logic to compute future control commands, and actuators execute commands to control the physical process (e.g., opening a valve). Sensors, actuators, and controllers operate in tandem to maintain a steady control loop of a physical process. When attacking an ICS, adversaries will manipulate a subset of sensor or actuator values to disrupt this control loop and cause physical harm. In this work, we leverage three popular ICS: Secure Water Treatment (SWaT) is a dataset of normal operations and accompanying anomalies from a water treatment plant (Goh et al., 2016); the Tennessee Eastman Process (TEP) is a simulation of a chemical process (Bathelt et al., 2015) which has been modified for anomaly simulation (Krotofil & Larsen, 2015; Fung et al., 2024); and CTown is a simulated water distribution network with an accompanying module for conducting cyberattacks (Murillo et al., 2021). A summary of the three ICS is provided in Table 1.

Unlike anomalies in other time-series domains, ICS anomalies are long-context (i.e., manipulations often occur over several consecutive datapoints), and involve highly interconnected interactions (i.e., when performed at the right time, even modifying a single sensor to an in-distribution value can cause harmful, cascading effects). For example, Fig. 1 shows the process-value manipulation for an attack provided in the SWaT dataset: the attacker spoofs a sensor value for 373 consecutive seconds to disrupt the industrial process. Since the attack is performed over a long duration and with an in-distribution value, it is difficult to detect with broad, statistical measures. Given the unique properties of ICS anomalies, we investigate whether time-series FMs are effective at detecting them.

*Table 1.* A summary of the datasets used for evaluation from three ICS (SWaT, TEP, and CTown). For each ICS, we show the number of features $d$, the number of samples in the training data $n$, the total time duration of the training data $t$, and the number of attacks in the test dataset.

| ICS | $d$ | Training Data $(n, t)$ | # Attacks |
|-----|-----|------------------------|-----------|
| SWaT | 51 | $n = 495000$, 7 days | 32 |
| TEP | 53 | $n = 96000$, 4 days | 22 |
| CTown | 39 | $n = 149760$, 14 days | 7 |

### 2.2. ICS anomaly detection

To detect ICS anomalies, many prior works train models with process data from normal ICS operations (Fung et al., 2022; Lamberts et al., 2023). Common architectures for ICS anomaly detection include one-dimensional CNNs (Kravchik & Shabtai, 2022) and LSTMs (Perales Gómez et al., 2020). These models $\mathcal{M}$ are trained for forecasting: based on a fixed-size context window of process values as input, $\mathcal{M}$ predicts the next observed state in the industrial process: $\hat{x}_t = \mathcal{M}(x_{t-1}, x_{t-2}, \ldots, x_{t-h})$ where $x_t$ is a vector that represents the full set of $d$ sensor and actuator values observed at time $t$. In order to use these forecasting models for anomaly detection, the mean absolute error (MAE) is computed between the model's prediction $\hat{x}_t$ and the observed state $x_t$, and a high MAE (i.e., exceeding a threshold $\tau$) indicates an anomaly: $y_t = \mathbb{I}\{\frac{\sum(|\hat{x}_t - x_t|)}{d} > \tau\}$

Using forecast-based errors, previous work achieves detection F1-scores of 0.85–0.89 on the SWaT dataset (Kravchik & Shabtai, 2022; Perales Gómez et al., 2020). However, obtaining the data necessary to train ML models to achieve such performance is often difficult in practice and can introduce distributional bias (Fung et al., 2025; Ahmed et al., 2020); thus, we investigate the application of time-series FMs to perform ICS anomaly detection without requiring training data.

## 3. Results

In this section, we describe the results of our preliminary investigation of FMs for ICS anomaly detection. We first demonstrate that pre-trained FMs and lightweight finetuning perform poorly with baseline detection strategies (Sec. 3.1 and Sec. 3.3), and demonstrate reasons for this poor performance with a case study (Sec. 3.2). We then propose and evaluate Forecast Fallback: a new forecasting method that improves ICS anomaly detection (Sec. 3.4).

### 3.1. Using baseline detection strategies with FMs

In this section, we first describe our evaluation of the baseline approach for using foundation models for ICS anomaly detection. We use two state-of-the-art, pre-trained,

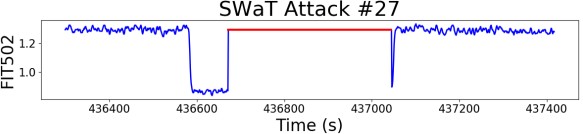

*Figure 1.* An example of a long-context ICS anomaly from the SWaT dataset: in SWaT attack #27, the attack is performed by replacing the value of sensor "FIT502" with an in-distribution value of 1.29 for 373 consecutive seconds.

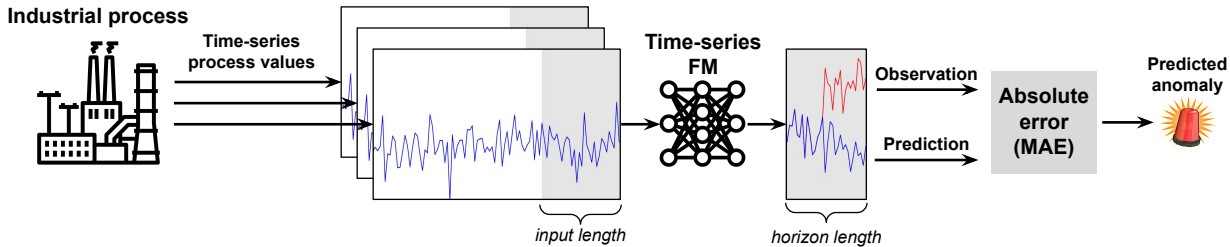

*Figure 2.* An overview of the baseline approach for applying time-series FMs for ICS anomaly detection: as input, the time-series foundation model receives a fixed input length of process values from an industrial process (e.g., sensor values for the past 32 seconds); as output, the time-series foundation model predicts a fixed output horizon length of process values (e.g., predicting the next 16 seconds of process values). Next, the prediction and true observation are compared, the mean absolute error (MAE) is computed, and high MAEs are used to predict anomalies.

foundation models for multivariate time-series forecasting: Chronos-2 (Ansari et al., 2024) and Moirai-2.0 (Liu et al., 2025). We adopt the detection strategy used in the majority of prior work in ML-based ICS anomaly detection (Fung et al., 2022; Lamberts et al., 2023): with a fixed history length (input window) of process data as input, use an ML model to forecast process values for a fixed output-time horizon (output window), and compute the mean absolute error (MAE) between the forecasted values and the ground truth. To provide a general, threshold-agnostic measure of goodness of fit, we report the Receiver Operating Characteristic Area Under the Curve (ROC-AUC). Figure 2 provides an overview of this baseline approach. Unlike prior work that must train a new ML model for time-series forecasting under each condition (e.g., different input length, different horizon length, different ICS), we are able to perform zero-shot forecasting (i.e, predictions without any training) with the same underlying FMs across multiple conditions.

In Appendix A, Table 2 provides the results of our baseline evaluation. Across all input-length and output-horizon configurations, the detection performance is poor, with very low ROC-AUC scores (i.e, between 0.39 and 0.50) that often fail to outperform the random-selection baseline of 0.5.

## 3.2. Case study: why baseline approaches fail

To help illustrate why baseline approaches are ineffective, we show example outcomes from the TEP dataset in Fig. 3; on the left half of Fig. 3 we show the Chronos-2 forecasted process value of the attacked feature (top) and the resulting MAE (bottom). The FM's forecast is highly accurate for both benign and anomalous data. Thus, when anomalous data is included in the input to Chronos-2, the model will reproduce these anomalous values in its forecast, resulting in a low overall MAE. Aside from an initial spike when the attack first begins, detection of long-context anomalies will fail when applying baseline approaches, particularly when input values are manipulated over a prolonged duration. To address this pitfall, we propose Forecast Fallback (shown

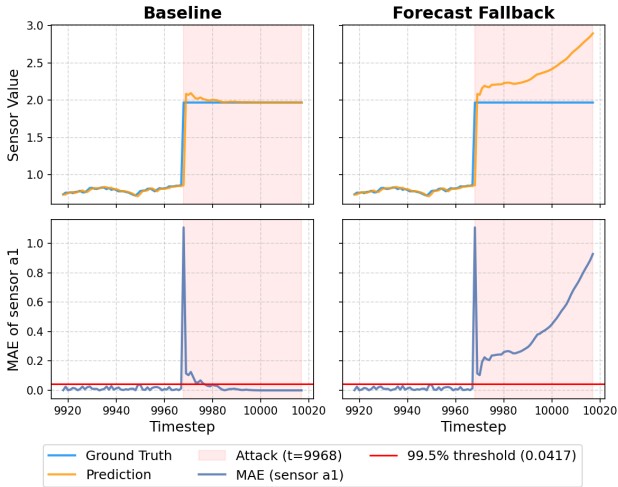

*Figure 3.* We show the ground truth and prediction of the attacker-manipulated feature (top) and the prediction's resulting MAE (bottom). On left, we show the results for the baseline detection approach: the prediction and the ground truth remain close even after the anomaly, leading to MAE values that are within the benign range and that fail to detect the anomaly over long duration. On right, we show the results with Forecast Fallback: the anomalous data is rejected as input in real-time and the MAE remains sufficiently high for detection.

on the right half of Fig. 3): a new forecasting strategy that rejects the intake of anomalous data and falls back on prior predictions, which is described in Sec. 3.4.

## 3.3. Does fine-tuning help?

A common approach for improving FM performance is to perform parameter-efficient fine-tuning (PEFT) with data that better represents the desired prediction domain (Zhang et al., 2025). In this section, we evaluate the effectiveness of PEFT with low-rank adaptation (LoRA) for ICS anomaly detection with FMs. We perform PEFT with LoRA (rank $r = 8$, scaling $\alpha = 16$, 500 steps) on Chronos-2 with various amounts of data from TEP: 2%, 10%, 50%, and

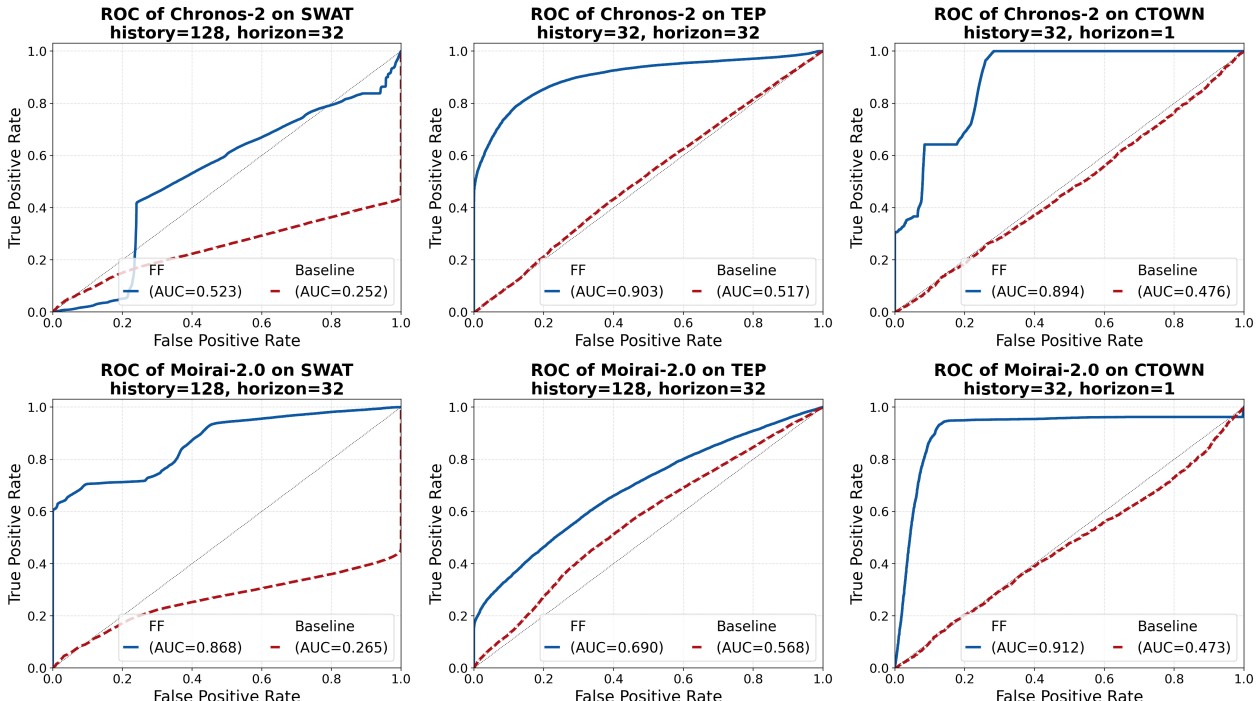

*Figure 4.* Improved ROC with FF across Chronos-2 (top) and Moirai-2.0 (bottom) models.

100% of the training dataset; in each case, we observe that the fine-tuning loss converges within 500 steps under the chosen parameters. We find that, regardless of the amount of data used, the forecasting ability of FMs only improves marginally: fine-tuning increases the ROC-AUC score from 0.517 to 0.519. We show the full results for our PEFT experiments in Table 3 in Appendix A.

### 3.4. Our proposed method: Forecast Fallback (FF)

Finally, we propose a new strategy to improve FM performance. Since FMs repeat the anomalous observations used in their input, we devise an approach that rejects the intake of suspicious data. Our approach, Forecast Fallback (FF), monitors prediction error in real-time. When predictions differ significantly from their observed counterparts, we stop trusting the observed data and instead use the FM's prediction as a surrogate input for that time step. We continue forecasting on the reconstructed input sequence to perform anomaly detection.

In our experiments, we use a 99.5th-percentile per-feature threshold on the MAE for FF and trigger the fallback process if three consecutive time-steps of a feature exceed this threshold. We leave further explorations for the best thresholding strategy for FF to future work; an FF-threshold that is too strict and fails to trigger FF will produce the same phenomenon shown in Fig. 3, whereas an FF-threshold that is too permissive will lead to stale predictions and false

positives from forecasts diverging from the observed data.

We illustrate the impact of FF on the right half of Fig. 3: the FM no longer reproduces the anomalous value as a part of its forecast. In Fig. 4, we show the ROC curves with and without FF for the best-performing configuration for each model on the SWaT, TEP, and CTown datasets. We find that FF improves detection performance across all configurations; for instance, the ROC-AUC on CTown improves from below 0.5 to over 0.89 for both Chronos-2 and Moirai-2.0. We determine the optimal detection threshold from each ROC curve and report the resulting F1 score with FF in Table 4 in Appendix A—0.74 for Chronos-2 and 0.83 for Moirai-2.0, which is competitive with the F1-scores of 0.7–0.9 reported for CTown in prior work (Erba et al., 2024).

## 4. Conclusion

Although time-series FMs have shown promise in several domains, we show that they perform poorly in detecting long-context ICS anomalies. In response, we propose a Forecast Fallback (FF) approach that rejects the intake of new, anomalous inputs and instead falls back to using previously forecasted process values as input when performing long-context forecasting; across three datasets and two foundation models, we show that FF significantly improves detection performance. Our preliminary results show promise for the application of foundation models for ICS, where training new ML models is often a challenge.

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

*Table 2.* We show the baseline ROC-AUC scores for each dataset and model, across different input lengths $I$ (along rows) and output-horizon lengths $H$. Across all configurations, the ROC-AUC is low (i.e., 0.2–0.6), indicating that FMs perform poorly off-the-shelf for ICS anomaly detection.

| | | SWAT ROC-AUC | | | TEP ROC-AUC | | | CTown ROC-AUC | | |
|---|---|---|---|---|---|---|---|---|---|---|
| | | $H = 1$ | $H = 16$ | $H = 32$ | $H = 1$ | $H = 16$ | $H = 32$ | $H = 1$ | $H = 16$ | $H = 32$ |
| **Chronos** | $I = 32$ | 0.239 | 0.241 | 0.251 | 0.463 | 0.492 | **0.517** | **0.476** | 0.443 | 0.396 |
| | $I = 64$ | 0.247 | 0.241 | 0.251 | 0.459 | 0.478 | 0.500 | 0.473 | 0.436 | 0.393 |
| | $I = 128$ | 0.249 | 0.244 | **0.252** | 0.461 | 0.474 | 0.496 | 0.458 | 0.439 | 0.402 |
| **Moirai** | $I = 32$ | 0.239 | 0.251 | 0.257 | 0.468 | 0.536 | 0.562 | **0.473** | 0.402 | 0.385 |
| | $I = 64$ | 0.247 | 0.253 | 0.259 | 0.470 | 0.541 | 0.567 | 0.459 | 0.430 | 0.376 |
| | $I = 128$ | 0.254 | 0.258 | **0.265** | 0.470 | 0.541 | **0.568** | 0.455 | 0.445 | 0.392 |

*Table 3.* ROC-AUC scores for finetuned Chronos-2 on TEP across different percentages of validation data used for training.

| Model | ROC-AUC by Fine-tuning Data Volume | | | |
|---|---|---|---|---|
| | **10%** | **20%** | **50%** | **100%** |
| Chronos-2 | 0.5197 | 0.5193 | 0.5191 | 0.5194 |

*Table 4.* Optimal MAE detection threshold F1 scores using Forecast Fallback approach on the optimal input and horizon lengths of each model and dataset.

| Model | F1 score using optimal thresholding | | |
|---|---|---|---|
| | **SWAT** | **TEP** | **CTOWN** |
| Chronos-2 | 0.248 | 0.756 | 0.742 |
| | $I = 128\ H = 32$ | $I = 32\ H = 32$ | $I = 32\ H = 1$ |
| Moirai-2.0 | 0.722 | 0.583 | 0.831 |
| | $I = 128\ H = 32$ | $I = 128\ H = 32$ | $I = 32\ H = 1$ |

# A. Full evaluation results

In this Appendix, we provide our full set of evaluation results from the experiments described in Sec. 3.

Table 2 shows the full set of ROC-AUC scores across three datasets (SWaT, TEP, and CTown), two foundation models (Chronos-2 and Moirai-2.0), and nine combinations of input length and output horizon length. Across all experiments, the ROC-AUC is below 0.6, indicating that the detection performance only slightly exceeds the 0.5 random baseline.

Table 3 shows the ROC-AUC scores after performing parameter-efficient fine-tuning (PEFT) with varying proportions of training data from TEP on Chronos-2, using an input length of 32 and an output horizon length of 32. Even when the full set of training data is used, there is no significant increase in ROC-AUC from PEFT.

Table 4 shows F1 scores using the optimal detection threshold and the Forecast Fallback (FF) method for Chronos-2 and Moirai-2.0 across three datasets: SWaT, TEP, and CTown. The input ($I$) and horizon ($H$) lengths correspond to the optimal settings identified in the baseline experiments in Table 2. These scores demonstrate that the improved ROC-AUC results translate to more effective anomaly detection performance

