# OpenReview forum: "On the Application of Time-Series Foundation Models for Detecting Long-Context Anomalies in Industrial Control Systems"
_ICML.cc/2026/Workshop/FMSD — FMSD @ ICML 2026 Poster_

### Official Review · Reviewer_jPPD · 2026-05-19
**Useful diagnostic study, but the methodological contribution is modest**

**Rating:** 6
**Confidence:** 4

**Review:**

### Summary

This paper studies whether off-the-shelf time-series foundation models can be used for anomaly detection in industrial control systems. The main finding is that Chronos-2 and Moirai-2.0 perform poorly on long-context ICS anomalies under standard forecasting-based detection. To address this, the paper proposes Forecast Fallback (FF), a simple heuristic that replaces suspicious observed inputs with prior model predictions, leading to improved ROC-AUC and F1 in the reported experiments.

### Strengths

1. The paper examines a relatively less explored application setting for time-series foundation models.

2. The negative result is useful: the paper identifies a concrete failure mode in which the forecasting model quickly absorbs anomalous observations and therefore loses detection sensitivity.

3. The proposed FF heuristic is simple and easy to understand, and the reported gains are nontrivial on the tested datasets.

4. The paper is generally clear about the practical motivation and the intended deployment scenario.

### Areas for Improvement

1. The main contribution is more diagnostic than methodological. FF is essentially a thresholded forecasting heuristic rather than a more substantial modeling advance.

2. The empirical scope is still limited, with only two TSFMs, three ICS datasets, and one general anomaly-scoring setup.

3. The method depends on thresholding choices, but the sensitivity analysis is limited.

4. The comparison to trained ICS-specific anomaly detectors is indirect, making it difficult to judge how competitive the approach really is.

### Detailed Comments

1. In my view, the most useful aspect of the paper is the failure analysis. It is informative to see that off-the-shelf TSFMs can forecast anomalous trajectories too faithfully once those trajectories enter the context window.

2. FF is a reasonable practical response to that issue, but I would suggest framing it more modestly. The current contribution is closer to a practical detection heuristic than to a new model or learning principle.

3. The paper would be stronger with a more systematic study of the FF trigger, especially the percentile threshold and the consecutive-step condition.

4. The fine-tuning analysis is also somewhat limited. Since it is only carried out on Chronos-2 and one dataset, I would be careful about making broader claims regarding the ineffectiveness of PEFT.

5. A direct comparison table against stronger ICS anomaly detection baselines under matched evaluation would make the empirical case easier to assess.

### Justification of Score

Overall, I think this is a useful empirical and diagnostic workshop submission. My main reservation is that the methodological contribution is relatively modest and the current validation scope is still limited. I would place it slightly above the borderline because the identified failure mode is relevant and the practical heuristic may still be of interest to the workshop audience.

---

### Official Review · Reviewer_c8Jc · 2026-05-20
**Insightful study but still preliminary**

**Rating:** 6
**Confidence:** 4

**Review:**

This paper investigates whether pre-trained time-series FMs (Chronos-2, Moirai-2.0) can detect anomalies in ICS without task-specific training. The authors identify a core failure mode — FMs reproduce anomalous inputs, suppressing the prediction error signal — and propose Forecast Fallback (FF), a real-time input sanitization strategy that substitutes previously predicted values for suspicious observations. Evaluated across three ICS datasets, FF improves ROC-AUC by 0.1–0.45 and achieves F1 scores competitive with trained ML baselines on CTown.

Strengths
1. Practically grounded problem. Zero-shot FM deployment directly addresses data scarcity and expertise constraints in ICS operations — well-suited to the workshop theme.
2. Clear failure-mode diagnosis. The Section 3.2 case study is the paper's strongest contribution: the FM's accurate reproduction of anomalous inputs collapses the MAE signal except at attack onset — a crisp, transferable insight.
3. FF delivers meaningful gains. CTown ROC-AUC improves from below 0.5 to over 0.89 for both models, with F1 scores competitive with prior supervised work.

Areas for Improvement
1. FF is underspecified. The 99.5th-percentile threshold and 3-timestep trigger are stated without ablation. Without a principled threshold-setting approach, deployability in real IC, where labeled attack data is unavailable, is unclear.
2. SWaT failure is unexplained. Chronos-2 achieves an F1 of only 0.248 on SWaT versus 0.742 on CTown. This substantial failure on the largest dataset receives no analysis.
3. Fine-tuning analysis is incomplete. PEFT is only evaluated on TEP with Chronos-2. The conclusion that fine-tuning does not help is generalized beyond what the evidence supports.
4. Inter-feature interactions are unaddressed. Despite identifying correlated sensor behavior as a defining ICS property, FF operates on per-feature thresholds independently.

Detailed Comments

Threshold calibration: A discussion of how FF thresholds would be set without labeled anomalies — e.g., using benign MAE distributions from a burn-in period — would substantially improve practical credibility.
SWaT analysis: Even a brief investigation of why FF underperforms on SWaT (feature count, attack diversity, input window length) would improve the paper's diagnostic value.
Baseline comparison: The claim that FF matches trained ML models rests entirely on CTown. F1 comparisons on SWaT and TEP against supervised baselines are absent.


Justification of Score
The failure-mode diagnosis is clear and well-supported, and FF is an intuitive fix with meaningful gains on two of three datasets. However, unjustified hyperparameters, an unexplained SWaT failure, and incomplete fine-tuning analysis limit the contribution's rigor.
Recommended Score: Borderline Accept
The core insight warrants workshop presentation. Acceptance recommended contingent on authors more explicitly acknowledging the SWaT failure and threshold-setting limitations.

---

### Official Review · Reviewer_dCNd · 2026-05-22
**A Strong Warning Against Naive TSFM Deployment**

**Rating:** 7
**Confidence:** 5

**Review:**

This paper asks a practical question: can off-the-shelf time-series foundation models detect cyber-physical anomalies in industrial control systems? The negative result is the most convincing part. Chronos-2 and Moirai-2.0, used with standard forecast-error scoring, perform poorly on SWaT, TEP, and CTown; many ROC-AUC values sit near or below chance. The diagnosis is plausible: during long attacks, anomalous readings enter the context window and the forecaster learns to reproduce them, so the residual signal disappears after an initial spike.

Forecast Fallback is a simple and interesting response. By rejecting suspicious observations and feeding previous predictions back into the model, it tries to stop anomalous context from contaminating future forecasts. The reported gains are large on some settings, including CTown, and the F1 results suggest the idea deserves follow-up.

Still, the method is preliminary. Raw MAE thresholding is a weak baseline for ICS anomaly detection, and optimal-threshold F1 can overstate deployable performance if thresholds are tuned with test labels. Forecast Fallback may also drift in open loop when normal processes change or attacks last a long time. The paper is valuable mainly as a warning about naive TSFM deployment in ICS, not as a finished detection system.